# ShadowFM: Geometric Approaches for Learning Quantum Many-Body States with Flow Matching on Classical Shadows

## Abstract

We introduce **ShadowFM**, a non-autoregressive flow matching for learning ground states of quantum many-body systems by learning distributions of classical shadows of a target quantum state with geometric considerations. Specifically, we conditionally generate shadows of an interested Hamiltonian's ground state by incorporating geometric modifications into the existing flow matching frameworks. We interpret the underlying geometry of shadows as a spherical manifold, inspired by the Bloch sphere, and motivate geometric approaches for modeling flow on the curved manifold. This approach enables us to capture the intrinsic symmetries of quantum measurements and allows more accurate sampling of Hamiltonian-conditioned shadows, which is a direction that was not explored in the previous works. In this work, we propose two methods, Riemannian-based and probability path-based method, to learn a more accurate transport dynamics on a Riemannian manifold equipped with the Bloch sphere geometry and on an anisotropic probability simplex, respectively. We demonstrate that this geometric consideration leads to more faithful sampling of shadows and leads to more accurate prediction of an unseen quantum state's observables, such as correlation functions and entanglement entropy.

## 1 Introduction

Flow Matching (FM) has recently emerged as a powerful generative modeling framework, achieving strong performance in both continuous (Lipman et al., 2022; Chen & Lipman, 2023a; Lipman et al., 2024; Albergo et al., 2023) and discrete (Stark et al., 2024; Gat et al., 2024; Davis et al., 2024; Cheng et al., 2024; 2025) domains. By learning the velocity field that transforms simple priors into complex data distributions, it achieves efficient sampling with simulation-free training. Extensions of flow matching to Riemannian manifolds have further allowed modeling data with non-Euclidean geometric structure, such as spherical or more complex geometries (Chen & Lipman, 2023b).

In parallel, quantum computers and quantum experiments have become larger and more sophisticated making it difficult to extract physical observables or results from their respective quantum state. As the full density matrix scales exponentially with system size, it is intractable to represent the quantum state using full tomography. To resolve this, classical shadow tomography (Huang et al., 2020) was proposed as a powerful protocol to represent the quantum state as a distribution of shadows, where the physical observables can be computed as an expectation of a function over shadows.

One of the promising directions in machine learning for quantum many-body physics is to leverage conditional generative models to learn a Hamiltonian-conditional distribution of classical shadows. If a generative model can generalize across a family of Hamiltonians using shadows obtained from simulations or quantum experiments, it can be leveraged to infer the ground state of both seen and unseen Hamiltonians from which properties of those ground states can be computed. As classical compute is cheap compared to performing quantum experiments, we can accurately estimate physical observables by generating an arbitrarily large number of shadows via inexpensive classical inference. For seen Hamiltonians, we effectively exchange stochastic yet unbiased errors, arising

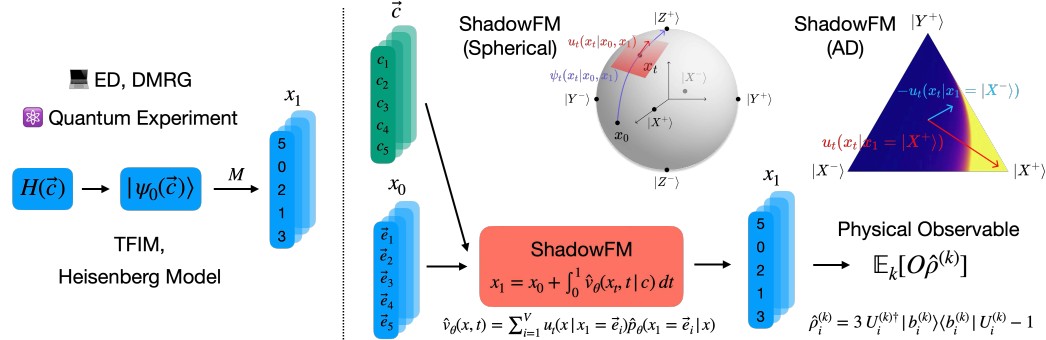

Figure 1: **Overview of ShadowFM.** Our generative models are trained on classical shadows $x_1$ generated from random single qubit measurements of the ground states $|\psi_0(\vec{c})\rangle$ of a class of Hamiltonians $\{H(\vec{c})\}$ with varying coupling constant $\vec{c}$. We classically simulate these states generating shadows from exact diagonalization or DMRG (Schollwöck, 2005). Then, we employ flow matching to learn the Hamiltonian conditonal-veocity field $\hat{v}_\theta(x_t, t|\vec{c})$ that transforms $x_0$ to $x_1$ along a path on a chosen geometry. Finally, we reconstruct local shadow estimators $\hat{\rho}_i^{(k)} = 3U_i^{(k)\dagger}|b_i^{(k)}\rangle\langle b_i^{(k)}|U_i^{(k)} - \mathbb{I}$ to estimate the expectation of physical observables as $\mathbb{E}_k[\mathrm{tr}(O\hat{\rho}_i^{(k)})]$.

from the finite number of classical shadows, for deterministic but biased errors introduced by the generative model's learned shadow distribution. For unseen Hamiltonians, we will be able to estimate observables that we could only otherwise evaluate by naive interpolation.

Recently, few works have explore training generative models to learn the distribution of shadows using autoregressive models (Carrasquilla et al., 2019; Yao & You, 2024), energy-based models (Jayakumar et al., 2024) and diffusion models (Tang et al., 2025). However, these methods either suffer from sequential bottlenecks of auto-regressiveness or do not respect the intrinsic geometric structure of shadows. To address these limitations, we develop a non-autoregressive, geometric Flow Matching method tailored specifically to generate shadows. Moreover, existing methods disregard the discrete nature and geometry of shadow, leaving significant untapped potential for improving performance if this property were explicitly exploited.

Based on the motivation to address these limitations, we introduce **ShadowFM**, a geometric flow matching framework for learning distribution of shadows, corresponding to a target quantum state. Building upon the well-established riemannian flow matching (RFM) (Chen & Lipman, 2023a) and continous state discrete flow matching (CS-DFM) (Cheng et al., 2025; 2024; Stark et al., 2024), we develop two geometric approaches: (1) *Riemannian approach* to model conditional velocity field and (2) *modification of conditional probability path* that respects the intrinsic geometry of shadows.

Our main contributions can be summarized as follows:

- We motivate incorporating geometric structure into generative modeling of classical shadows, drawing inspiration from the Bloch sphere representation. We show experiments that demonstrate respecting this underlying geometry significantly improves the accuracy of observable estimation.

- We develop two geometric approaches to design geometric flow matching specifically for learning shadows: (1) a *Riemannian manifold*-based method and (2) an *anisotropic probability path*-based method, which generalizes existing CS-DFM. These are motivated from the Bloch sphere geometry and the inherent (target, anti-target) pairing structure of shadows.

- We empirically evaluate our method's efficacy on transverse-field Ising model and Heisenberg model, comparing to various classical machine learning models and non-autogressive baselines.

## 2 BACKGROUND

### 2.1 CLASSICAL SHADOW REPRESENTATION OF QUANTUM STATES

A central challenge in quantum information is the exponential complexity of representing quantum states: the density matrix of an $n$-qubit system requires $O(2^{2n})$ parameters, making direct storage

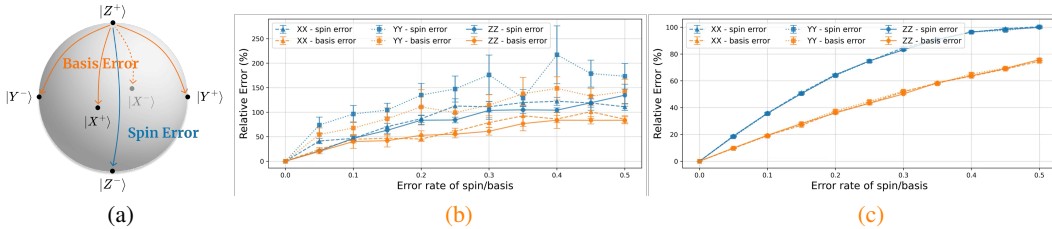

(a)        (b)        (c)

Figure 2: **(a) Illustration of two error types valid in shadow. (b),(c) Effect of measurement error rate on the reconstruction of physical observables for TFIM ($L = 6$) and Heisenberg model ($L = 6$).** Notice that a spin error, which takes us across the Bloch sphere, is more detrimental across all error rate and leads to significantly higher error in reconstructing observable. These values are averaged over 100 ground states of Hamiltonians.

and computation intractable for even moderately large $n$, *e.g.*, $n \approx 30$. Classical shadow tomography (Huang et al., 2020) provides an efficient encoding of a quantum state into an ensemble of randomized measurement outcomes, or shadows. These shadows serve as succint surrogates of the full density matrix while retaining the ability to accurately predict a wide range of observables efficiently.

The Pauli-6 positive operator-valued measure (POVM) performs measurements in the eigenbases of the three Pauli operators. Each single-qubit measurement yields a label $s \in \{|X^+\rangle, |X^-\rangle, |Y^+\rangle, |Y^-\rangle, |Z^+\rangle, |Z^-\rangle\}$ that indicates the pair of basis it was measured in and the output of the measurement. In algorithm 1 and algorithm 2, the protocol to acquire shadows and to estimate observables is explained. Refer to section A, for more details.

## 2.2 FLOW MATCHING

Flow matching (FM) (Lipman et al., 2022; Liu et al., 2022; Albergo et al., 2023) is a generative modeling framework that learns velocity fields transporting prior distributions to target data distributions, with simulation-free training. The linear flow matching framework trains a velocity field predictor $v_\theta$ with a conditional flow matching (CFM) loss:

$$\mathcal{L}_{\text{CFM}}(\theta) = \mathbb{E}_{t\sim U[0,1],\, x_1\sim q(\cdot),\, x_1\sim p_t(\cdot|x_1)} \|v_\theta(t, x_t) - u_t(x_t \mid x_1)\|_2^2, \tag{1}$$

where $q$ is target distribution, $p_t(\cdot|x_1)$ is the conditional probability path, and $u_t(x_t|x_1)$ is the conditional velocity field which is usually defined from linear interpolant $(x_1 - x_t)/(1 - t)$. Upon convergence, one can solve ODE $\frac{dx_t}{dt} = \hat{v}_\theta(x_t, t)$, to generate samples from the target distribution.

## 2.3 RIEMANNIAN FLOW MATCHING

The key idea in Riemannian Flow Matching (RFM) (Chen & Lipman, 2023a) is to extend Flow Matching to manifolds with curved geometry. Given two points $x_0, x_1 \in \mathcal{M}$, where $\mathcal{M}$ is a smooth Riemannian manifold endowed with metric tensor $g$, the exponential map $\exp_{x_0} : T_{x_0}\mathcal{M} \to \mathcal{M}$ projects a tangent vector $v \in T_{x_0}\mathcal{M}$ to the manifold along a geodesic starting at $x_0$ in the direction of $v$. Conversely, the logarithmic map $\log_{x_0} : \mathcal{M} \to T_{x_0}\mathcal{M}$ returns the initial velocity vector that generates a geodesic from $x_0$ to another point $x_1$.

If $\mathcal{M}$ has the closed-form expressions for $\exp$ and $\log$ maps, we can define a geodesic interpolation $x_t$ between $x_0$ and $x_1$ as $x_t = \psi_t(x_0) = \exp_{x_0}\left(t \log_{x_0}(x_1)\right)$. This yields the conditional velocity field $u_t(x_t|x_0, x_1) = \frac{d}{dt}x_t = \log_{x_t}(x_1)/(1 - t)$, which satisfies the ODE $\frac{d}{dt}\psi_t(x) = u_t(\psi_t(x))$, and is used as the regression target in the RFM objective:

$$\mathcal{L}_{\text{RFM}} = \mathbb{E}_{t\sim U[0,1], x_0\sim p_0(\cdot), x_1\sim q(\cdot)} \left[\|v_\theta(x_t, t) - u_t(x_t|x_0, x_1)\|_g^2\right]. \tag{2}$$

# 3 METHOD

## 3.1 MOTIVATION: BLOCH SPHERE GEOMETRY OF QUANTUM STATES

Bloch sphere (Bloch, 1946; Feynman et al., 1957; Nielsen & Chuang, 2010) is a natural geometric representation of quantum states. We first motivate this with a simple experiment by considering

how strongly errors in the shadow data affect the accuracy of estimating the physical observable in Section 3.1.

**Shadows live on $\mathbb{CP}^1$.** A pure quantum state is a ray in $\mathbb{C}^2$, *i.e.*, $|\psi\rangle \sim e^{i\gamma}|\psi\rangle$ for any global phase $\gamma$. The space of rays is the complex projective line: $\mathbb{CP}^1 = (\mathbb{C}^2 \setminus \{0\})/\sim$, $(z_0, z_1) \sim \lambda(z_0, z_1)$, $\lambda \in \mathbb{C} \setminus \{0\}$. Hence, each pure qubit or single qubit shadow corresponds to a point $[z_0 : z_1] \in \mathbb{CP}^1$.

**Equivalence between Fubini–Study metric on $\mathbb{CP}^1$ and the natural metric on $S^2$.** The Fubini-Study metric is the canonical, unitary-invariant metric on complex projective space $\mathbb{CP}^n$. For a normalized ket $|\psi\rangle$ and tangent variation $|d\psi\rangle$ orthogonalized by phase, using the Bloch sphere parametrization $|\psi(\theta, \phi)\rangle = \cos\frac{\theta}{2}|0\rangle + e^{i\phi}\sin\frac{\theta}{2}|1\rangle$, its line element can be written as the following:

$$ds^2_{\text{FS}} = \langle d\psi \mid d\psi \rangle - |\langle \psi \mid d\psi \rangle|^2 = \tfrac{1}{4}\big(d\theta^2 + \sin^2\theta\, d\phi^2\big).$$

Note that there is a smooth bijection (the Bloch map) between $\mathbb{CP}^1$ and the 2-sphere $S^2 \equiv \{x \in \mathbb{R}^3 \mid \|x\|_2^2 = 1\}$ given by $[z_0 : z_1] \mapsto \boldsymbol{n} = \langle\psi|\boldsymbol{\sigma}|\psi\rangle \in S^2$. Since the natural metric on the unit sphere $S^2$ is: $ds^2_{S^2} = d\theta^2 + \sin^2\theta\, d\phi^2 = 4ds^2_{\text{FS}}$, $\mathbb{CP}^1$ and $S^2$ are diffeomorphic and the metrics coincide up to a constant scale: Fubini-Study metric equals the natural metric on a sphere of radius $\frac{1}{2}$ and the Bloch map is an isometry up to a constant scale. Therefore, each single qubit state can be represented as a point on the three-dimensional unit ball with the bases as represented in fig. 2. This geometry is referred to as the *Bloch Sphere*.

**Toy experiment on the effect of shadow error on estimating observables.** To illustrate the affect of the Bloch sphere geometry, we conduct a simple experiment to simulate how probablistic errors in shadows affect the accuracy of estimating quantum observables. Specifically, in Figure 2, we report the RMSE of XX and ZZ correlation ($\langle\sum_{i=1}^{L-1}\sigma_i\sigma_{i+1}\rangle$, $\sigma \in \{X, Z\}$) as a function of the shadow error rate, using 100 ground states of the 6-qubits TFIM at various coupling constants (see Section 4.1 for the definition of TFIM). For shadows, there can be two types of errors: *spin error*, which flips the measured eigenvalue, *e.g.*, $|X^+\rangle \to |X^-\rangle$, and *basis error*, which rotates the measurement axis, *e.g.*, $|X^+\rangle \to |Y^\pm\rangle$ or $|Z^\pm\rangle$. We find that spin error leads to significantly higher reconstruction error in both metrics compared to basis error. This observation highlights that spin errors, which invert measurement outcomes within the same basis, are more detrimental to accurately estimating physical observables.

This result motivates us to design embeddings for shadows such that measurement outcomes differing by a spin flip are placed further apart in the embedding space compared to those differing by a basis flip. From this perspective, we design a geometric flow matching to suppress spin errors by formulating a flow that respects the geometry of Bloch sphere representation.

### 3.2 GEOMETRIC FLOW MATCHING FOR GENERATING SHADOWS

We introduce our two geometric approaches: (1) *Spherical Flow*, a Riemannian-based Flow Matching, that respects the $S^2$ geometry arising from the Bloch map in Section 3.2.1 and (2) *Anisotropic Dirichlet Flow*, a probability path-based Flow Matching, that respects the asymmetric structure of shadows in Section 3.2.2.

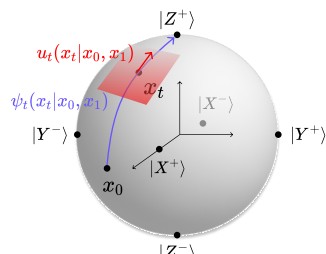

### 3.2.1 RIEMANNIAN APPROACH: SPHERICAL FLOW

We introduce our *Spherical flow*, a Riemannian-based approach for generating shadows, reflecting the spherical geometry arising from the Bloch map. Our contribution is to recognize that Pauli shadows naturally reside on the sphere $S^2$, and to apply Riemannian Flow Matching (RFM) to learn their distribution directly on this manifold.

Figure 3: **Illustration of the embedding of shadows on $S^2$ and our Spherical Flow.**

Formally, the geodesic path on $S^{K-1}$ can be calculated with the closed form of exponential map $\exp_p$ at a point $p \in S^{K-1}$ for a tangent vector $v \in T_p S^{K-1}$ and the logarithmic map $\log_p(q)$ at a

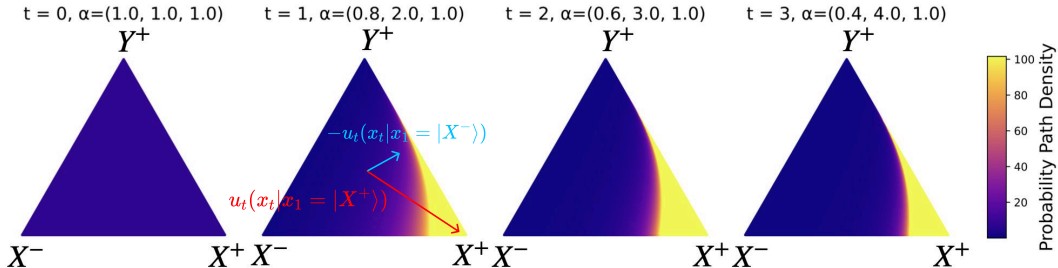

Figure 4: **Illustration of our Anisotropic Dirichlet Flow.** To visualize the effect of our Anisotropic Dirichlet flow, we present a $\Delta^2$ example in Figure 4. This illustrates the evolution of the anisotropic probability path over time $t$ (probabilities exponentiated for visual aid). Each vertex of the triangle represents distinct shadow: the left vertex corresponds to $|X^-\rangle$, the right to $|X^+\rangle$, and the top to a different Pauli basis (e.g., $|Y^+\rangle, |Y^-\rangle, |Z^+\rangle, |Z^-\rangle$). As $t$ increases, the density becomes sharply concentrated near the $|X^+\rangle$ vertex, while simultaneously repelling from $|X^-\rangle$.

point $p$ with respect to another point $q \in S^{K-1}$. These are given by:

$$\exp_p(v) = \cos(|v|)\, p + \sin(|v|)\, \frac{v}{|v|}, \qquad \log_p(q) = \frac{\cos^{-1}(\langle p, q\rangle)}{\sqrt{1 - \langle p, q\rangle^2}}\, (q - \langle p, q\rangle p). \tag{3}$$

where $\langle \cdot, \cdot \rangle$ denotes the standard inner product. In our shadow generation framework, we are interested in $K = 3$, hence the geometry of $S^2$. These maps facilitate efficient computation of geodesic paths on the $S^2$ sphere, enabling flow-based generative modeling within the spherical manifold framework.

For noise distribution, motivated from Cheng et al. (2024), we utilize a pushforward of the mapping $\pi : C^3 \equiv \{ x \in \mathbb{R}^3 \mid \|x\|_1 = 1 \} \to S^2$, $\pi_i : x_i \mapsto \mathrm{sgn}(x_i) \odot \sqrt{|x_i|}$, where $C^n$ is the $n$-dimensional cross polytope and $\odot$ denotes element-wise multiplication. For training, we optimize the Hamiltonian-conditional denoising classifier $\hat{p}_\theta(x_1|x_t, c)$ with cross entropy loss (equation 5):

$$\mathcal{L}_{\mathrm{CE}, S^2} = \mathbb{E}_{t\sim U[0,1], c\sim\mathcal{C}, x_0\sim\pi_*p_0(\cdot), x_1\sim q(\cdot|c)} \left[ -\log \hat{p}_\theta(x_1|x_t, c) \right], \text{ with } x_t = \exp_{x_0}(t \log_{x_0}(x_1)), \tag{4}$$

where $c$ denotes coupling constant of Hamiltonians (see Section 4) and $\pi_*$ is the pushforward of $\pi$. $p_0$ is the uniform distribution supported on the cross polytope $C^3$, which generalizes the uniform distribution on the 2-simplex $\Delta^2$ (see definition at Section 3.2.2), is defined as $\mathrm{Dir}(1,1,1) \times \mathrm{Unif}\{(\pm 1, \pm 1, \pm 1)\}$.

For inference, we can solve the Hamiltonian-conditional ODE $\frac{d}{dt}x_t = \hat{v}_\theta(x_t, t|\mathbf{c})$ by following the path on the spherical manifold $\hat{v}_\theta(x_t, t) = \sum_{i=1}^{K} u_t(x_t|x_1 = e_i)\hat{p}_\theta(x_1 = e_i|x_t)$, where $u_t(x_t|x_1 = e_i) = \frac{\log_{x_t}(x_1 = e_i)}{1-t}$ is the conditional velocity field on spherical manifold, to generate shadows.

### 3.2.2 PROBABILITY PATH APPROACH: ANISOTROPIC DIRICHLET FLOW

**Dirichlet Flow.** Dirichlet flow (Stark et al., 2024), as one of the flow matching method for discrete data, first pre-define a probability path with Dirichlet distributions $p_t(x \mid x_1 = \vec{e}_i) = \mathrm{Dir}(x; \alpha = \vec{1} + t\vec{e}_i)$ and the corresponding conditional velocity field $u_t(x \mid x_1 = \vec{e}_i)$. Then it trains a denoising classifier using a cross-entropy (CE) loss:

$$\mathcal{L}_{\mathrm{CE}}(\theta) = \mathbb{E}_{t\sim U[0,1], x_1\sim q(\cdot), x\sim p_t(\cdot|x_1)} \left[ -\log \hat{p}_\theta(x_1|x) \right]. \tag{5}$$

At inference time, one can construct the marginal velocity field by a linear combination of conditional velocity fields $u_t(x_t|x_1)$: $\hat{v}_\theta(x_t, t) = \sum_{i=1}^{K} u_t(x_t|x_1 = \hat{e}_i)\hat{p}_\theta(x_1 = \hat{e}_i|x)$, where $K - 1$ is the dimension of simplex. Solving the ODE with this marginal velocity field allows more stable generation path on the probability simplex, which is the manifold discrete data resides on.

**Anisotropic Dirichlet Flow.** We propose our *Anisotropic Dirichlet flow* as a generalization of Dirichlet flow, by utilizing modified probability path based Flow Matching that could incorporate inherent (target, anti-target) pairing structure of shadows.

We first pre-define the conditional probability path that induces anisotropic flow; a flow that pushes toward a target, but simultaneously pull away from an anti-target. Let $\Delta^{K-1} = \{x \in \mathbb{R}_{\geq 0}^K \mid \sum_{j=1}^K x_j = 1\}$ be the $(K-1)$-simplex with barycentric coordinates $x = (x_1, \ldots, x_K)$. Here, $K - 1$ is the dimension of simplex, *e.g.*, in our shadow generation framework, $K = 6$. For a fixed drift coefficient $\gamma \in [0, 1)$ and $t \in [0, 1]$, define the conditional probability path:

$$p_t^{\text{AD}}(\mathbf{x} \mid x_1 = e_i) = \frac{1}{Z(t)} \prod_{j=1}^K x_j^{\alpha_j(t)-1}, \qquad \alpha_j(t) = \begin{cases} 1 + t, & j = i, \\ 1 - \gamma t, & j = \bar{i}, \\ 1, & j \in \{1, \ldots, K\} \setminus \{i, \bar{i}\}. \end{cases}$$

(6)

with normalizer $Z(t) = \prod_{j=1}^K \Gamma(\alpha_j(t))/\Gamma(\alpha_\Sigma(t))$, where $\alpha_\Sigma(t) = \sum_{j=1}^K \alpha_j(t) = K + (1-\gamma)t$, and $\bar{i}$ denotes the conjugate pair of $i$. In our implementation we choose the pair as $(0, 1), (2, 3), (4, 5)$, representing pairs of shadows $(|X^+\rangle, |X^-\rangle), (|Y^+\rangle, |Y^-\rangle), (|Z^+\rangle, |Z^-\rangle)$, respectively. Intuitively, this probability path $p_t$ starts at the uniform Dirichlet distribution $\text{Dir}(1, \ldots, 1)$ and continuously tilts probability mass from the first vertex to the second as $t$ grows ($\dot{\alpha}_i = +1 \; \dot{\alpha}_{\bar{i}} = -\gamma$).

To find the conditional velocity field $u_t$ such that satisfies the continuity equation $\partial_t p_t + \nabla \cdot (p_t u_t) = 0$ for the pre-defined conditional probability path, we take an ansatz of projected-linear field that only acts along the two varying coordinates:

$$u_t^{\text{AD}}(x_t|x_1 = e_i) = \underbrace{C(x_i, t)(e_i - x_t)}_{\text{push toward } e_i} - \underbrace{\gamma D(x_{\bar{i}}, t)(e_{\bar{i}} - x_t)}_{\text{pull away from } e_{\bar{i}}},$$

(7)

where $e_i \in \mathbb{R}^K$ denotes the canonical basis vectors and $\gamma \in \mathbb{R}$ denotes the hyperparameter which controls the magnitude of pulling term. We set this to $\gamma = 0.1$ in the experiments. Intuitively, first term push towards the *target* (*e.g.* $|X^+\rangle$), while the second term pulls away from the *anti-target* (*e.g.* $|X^-\rangle$). Hence, this is a general framework that can be applied to model distribution of data which have pairs of target and anti-target. In section B, we explicitly derive $C(x_i, t)$ and $D(x_{\bar{i}}, t)$ by solving the continuity equation:

$$C(x_i, t) = -x_i^{-\alpha_i(t)+1} (1 - x_i)^{-\alpha_\Sigma(t)+\alpha_i(t)} \int_0^{x_i} s^{\alpha_i(t)-1} (1-s)^{\alpha_\Sigma(t)-\alpha_i(t)} \frac{\log s - \langle \log X_i \rangle_t}{1 - s} \, ds,$$

(8)

$$D(x_{\bar{i}}, t) = -x_{\bar{i}}^{-\alpha_{\bar{i}}(t)+1} (1 - x_{\bar{i}})^{-\alpha_\Sigma(t)+\alpha_{\bar{i}}(t)} \int_0^{x_{\bar{i}}} s^{\alpha_{\bar{i}}(t)-1} (1-s)^{\alpha_\Sigma(t)-\alpha_{\bar{i}}(t)} \frac{\log s - \langle \log X_{\bar{i}} \rangle_t}{1 - s} \, ds,$$

(9)

where $\langle \log X_i \rangle_t \equiv \mathbb{E}_{x_i \sim p_t}[\log x_i] = \psi(\alpha_i) - \psi(\alpha_\Sigma)$, and $\psi(z) = \frac{d}{dz} \log(\Gamma(z))$ is digamma function. Define $\beta(t) = S(t) - \alpha_2(t) = K - 1 - \gamma t$ and let $I_x(a, b)$ be the regularized incomplete Beta function, $B(a, b)$ the Beta function. If $\gamma = 0$, eq. (8) returns to $C(x_i, t) = -\tilde{I}_{x_i}(t + 1, K - 1) \frac{B(t+1, K-1)}{(1-x_i)^{K-1} x_i^t}$ and hence we return to $u_t(x|x_1 = e_i) = C(x_i, t)(e_i - x)$ as in Stark et al. (2024). Therefore, this can be viewed as the generalized Dirichlet flow, where we allow anisotropy to model flow that *push to target, pull away from anti-target*.

For training, one can similarly train the Hamiltonian conditional denoising classifer with cross-entropy loss but with anisotropic conditional probability path:

$$\mathcal{L}_{\text{CE, AD}}(\theta) = \mathbb{E}_{t \sim U[0,1], c \sim \mathcal{C}, x_1 \sim q(\cdot|\mathbf{c}), x_t \sim p_t^{\text{AD}}(\cdot|x_1)} \left[ -\log \hat{p}_\theta(x_1|x_t, c) \right].$$

(10)

For inference, one can then construct the marginal velocity field with the linear combination of conditional velocity fields: $\hat{v}_\theta(x_t, t|c) = \sum_{i=1}^K u_t^{\text{AD}}(x_t|x_1 = \hat{e}_i)\hat{p}_\theta(x_1 = \hat{e}_i|x_t, c)$, and then solve the ODE.

| Category | Method | RMSE (Correlation) ↓ | | | RMSE (Entropy) ↓ | | |
|---|---|---|---|---|---|---|---|
| | | $1k$ | $10k$ | $100k$ | $1k$ | $10k$ | $100k$ |
| Exact | CS | $0.086 \pm 0.000$ | $0.027 \pm 0.000$ | $0.008 \pm 0.000$ | $0.063 \pm 0.000$ | $0.022 \pm 0.000$ | $0.008 \pm 0.000$ |
| Classical | RBFK | | $0.028 \pm 0.001$ | | | $0.050 \pm 0.001$ | |
| | NTK | | $0.396 \pm 0.011$ | | | $0.064 \pm 0.002$ | |
| CFM | LinearFM | $0.212 \pm 0.016$ | $0.175 \pm 0.018$ | $0.170 \pm 0.019$ | $0.280 \pm 0.011$ | $0.297 \pm 0.013$ | $0.299 \pm 0.013$ |
| | Diff-LM | $0.188 \pm 0.014$ | $0.140 \pm 0.018$ | $0.134 \pm 0.019$ | $0.577 \pm 0.010$ | $0.564 \pm 0.012$ | $0.550 \pm 0.013$ |
| CS-DFM | StatisticalFM | $0.169 \pm 0.011$ | $0.133 \pm 0.014$ | $0.126 \pm 0.015$ | $0.164 \pm 0.005$ | $0.161 \pm 0.007$ | $0.164 \pm 0.007$ |
| | **Ours (Spherical)** | $0.103 \pm 0.005$ | $0.053 \pm 0.006$ | $0.041 \pm 0.007$ | $0.059 \pm 0.006$ | $0.049 \pm 0.006$ | $0.047 \pm 0.006$ |
| | **Ours (AD)** | $\mathbf{0.088 \pm 0.002}$ | $\mathbf{0.034 \pm 0.001}$ | $\mathbf{0.021 \pm 0.001}$ | $\mathbf{0.056 \pm 0.005}$ | $\mathbf{0.048 \pm 0.007}$ | $\mathbf{0.045 \pm 0.007}$ |

Table 1: **Quantiative comparison on 1D TFIM (L=10).**

| Category | Method | RMSE (Correlation) ↓ | | | RMSE (Entropy) ↓ | | |
|---|---|---|---|---|---|---|---|
| | | $1k$ | $10k$ | $100k$ | $1k$ | $10k$ | $100k$ |
| Exact | CS | $0.055 \pm 0.014$ | $0.024 \pm 0.004$ | $0.008 \pm 0.002$ | $0.054 \pm 0.000$ | $0.016 \pm 0.000$ | $0.008 \pm 0.000$ |
| Classical | RBFK | | $0.118 \pm 0.008$ | | | $0.080 \pm 0.004$ | |
| | NTK | | $0.436 \pm 0.018$ | | | $0.079 \pm 0.005$ | |
| CFM | LinearFM | $0.242 \pm 0.007$ | $0.160 \pm 0.008$ | $0.155 \pm 0.008$ | $0.280 \pm 0.007$ | $0.172 \pm 0.006$ | $0.170 \pm 0.006$ |
| | Diff-LM | $0.275 \pm 0.007$ | $0.210 \pm 0.007$ | $0.203 \pm 0.004$ | $0.370 \pm 0.007$ | $0.294 \pm 0.007$ | $0.286 \pm 0.007$ |
| CS-DFM | StatisticalFM | $0.166 \pm 0.006$ | $0.124 \pm 0.007$ | $0.120 \pm 0.007$ | $0.194 \pm 0.007$ | $0.128 \pm 0.009$ | $0.125 \pm 0.001$ |
| | **Ours (Spherical)** | $0.161 \pm 0.005$ | $0.124 \pm 0.007$ | $0.153 \pm 0.007$ | $\mathbf{0.104 \pm 0.009}$ | $\mathbf{0.073 \pm 0.008}$ | $\mathbf{0.069 \pm 0.008}$ |
| | **Ours (AD)** | $\mathbf{0.153 \pm 0.003}$ | $\mathbf{0.114 \pm 0.004}$ | $\mathbf{0.109 \pm 0.004}$ | $0.160 \pm 0.013$ | $0.105 \pm 0.012$ | $0.101 \pm 0.012$ |

Table 2: **Quantiative comparison on 1D TFIM (L=30).**

## 4 EXPERIMENTS

### 4.1 TRANSVERSE FIELD ISING MODEL (TFIM)

We consider the 1D spin-$\frac{1}{2}$ anti-ferromagnetic transverse-field Ising model (TFIM) on a chain of length $L$ with periodic boundary condition:

$$H_{\text{TFIM}}(c) = -(1 - c) \sum_{i=1}^{L} \sigma_i^z \sigma_{i+1}^z - c \sum_{i=1}^{L} \sigma_i^x, \tag{11}$$

where $c \in [0, 1]$ is a parameter that controls the strength of the transverse-field. TFIM is one of the simplest quantum many-body Hamiltonians that captures rich nontrivial physics, such as phase transition at the critical point $c = 1/2$.

We evaluate on accuracy of predicting two observables: two-point correlation function and entanglement entropy. We estimate the two observables for every pair of sites $(i, j)$, with $i, j \in \{1, \ldots, L\}$, and compute the root mean square error (RMSE) against the true expectation of observables for all pairs. The reported RMSE is averaged over a test set of 100 ground states of the Hamiltonian. See section C for the definition of two-point correlation fucntion and entanglement entropy.

In the following experiments, we compare our methods with various baselines. CS refers to the oracle classical shadow protocol from (Huang et al., 2020), which directly estimates observables from true shadows using, is an exact method which serves as an oracle reference. For classical machine learning baseline, we compare with kernel ridge regression based on radial basis function (RBFK) and neural tangent kernel (NTK) following Huang et al. (2022). We also compare with various flow matching and diffusion baselines including LinearFM (Lipman et al., 2022), Diff-LM (Li et al., 2022; Tang et al., 2025), StatisticalFM (Cheng et al., 2024; Davis et al., 2024). CFM refers to continuous flow matching and CS-DFM (Cheng et al., 2025) refers to continuous state discrete flow matching. For our AD flow, we evaluate for $\gamma \in \{0, 0.05, 0.1\}$ and report the best value. Refer to section D for detailed experimental settings.

**Learning ground states of TFIM.** We test generative models for $L = 10$ and $L = 30$, which are trained with shadows obtained from using exact diagonalization and DMRG (Schollwöck, 2005), respectively. We report the accuracy of shadow generation by estimating observables using $M_{\text{infer}} \in \{1k, 10k, 100k\}$ generated shadows across various methods.

**Phase Transition of TFIM.** In fig. 5, we present comparison of estimation of mean of ZZ correlation $\frac{1}{L} \sum_{i=1}^{L} \langle Z_i Z_{i+1} \rangle$ and mean of entanglement entropy $\frac{1}{L} \sum_{i=1}^{L} E_{i,i+1}$, where we are assuming

| CATEGORY | METHOD | RMSE (CORRELATION) ↓ | | | RMSE (ENTROPY) ↓ | | |
|---|---|---|---|---|---|---|---|
| | | $1k$ | $10k$ | $100k$ | $1k$ | $10k$ | $100k$ |
| Exact | CS | $0.050 \pm 0.005$ | $0.016 \pm 0.002$ | $0.005 \pm 0.001$ | $0.091 \pm 0.000$ | $0.017 \pm 0.000$ | $0.005 \pm 0.000$ |
| Classical | RBFK | | $0.076 \pm 0.002$ | | | $0.150 \pm 0.006$ | |
| | NTK | | $0.059 \pm 0.002$ | | | $0.118 \pm 0.004$ | |
| CFM | LinearFM | $0.083 \pm 0.002$ | $0.066 \pm 0.002$ | $0.065 \pm 0.002$ | $0.120 \pm 0.002$ | $0.086 \pm 0.002$ | $0.085 \pm 0.002$ |
| | Diff-LM | $0.147 \pm 0.002$ | $0.118 \pm 0.002$ | $0.116 \pm 0.002$ | $0.181 \pm 0.002$ | $0.150 \pm 0.002$ | $0.148 \pm 0.002$ |
| CS-DFM | StatisticalFM | $0.074 \pm 0.002$ | $0.056 \pm 0.002$ | $0.054 \pm 0.002$ | $0.125 \pm 0.002$ | $0.078 \pm 0.003$ | $0.076 \pm 0.003$ |
| | **Ours (Spherical)** | $\mathbf{0.066 \pm 0.002}$ | $\mathbf{0.044 \pm 0.002}$ | $\mathbf{0.042 \pm 0.002}$ | $\mathbf{0.115 \pm 0.002}$ | $\mathbf{0.059 \pm 0.003}$ | $\mathbf{0.055 \pm 0.004}$ |
| | **Ours (AD)** | $0.071 \pm 0.002$ | $0.049 \pm 0.002$ | $0.046 \pm 0.002$ | $0.124 \pm 0.003$ | $0.065 \pm 0.003$ | $0.060 \pm 0.003$ |

Table 3: **Quantiative comparison on 1D Heisenberg model (L=10).**

| CATEGORY | METHOD | RMSE (CORRELATION) ↓ | | | RMSE (ENTROPY) ↓ | | |
|---|---|---|---|---|---|---|---|
| | | $1k$ | $10k$ | $100k$ | $1k$ | $10k$ | $100k$ |
| Exact | CS | $0.055 \pm 0.005$ | $0.027 \pm 0.002$ | $0.008 \pm 0.001$ | $0.059 \pm 0.000$ | $0.030 \pm 0.000$ | $0.009 \pm 0.000$ |
| Classical | RBFK | | $0.072 \pm 0.001$ | | | $0.199 \pm 0.006$ | |
| | NTK | | $0.066 \pm 0.001$ | | | $0.168 \pm 0.003$ | |
| CFM | LinearFM | $0.146 \pm 0.001$ | $0.101 \pm 0.001$ | $0.111 \pm 0.001$ | $0.179 \pm 0.005$ | $0.194 \pm 0.006$ | $0.215 \pm 0.006$ |
| | Diff-LM | $0.146 \pm 0.001$ | $0.098 \pm 0.001$ | $0.111 \pm 0.001$ | $0.177 \pm 0.005$ | $0.176 \pm 0.006$ | $0.215 \pm 0.006$ |
| CS-DFM | StatisticalFM | $0.130 \pm 0.001$ | $0.079 \pm 0.001$ | $0.090 \pm 0.001$ | $0.154 \pm 0.005$ | $0.177 \pm 0.005$ | $0.182 \pm 0.005$ |
| | **Ours (Spherical)** | $\mathbf{0.105 \pm 0.001}$ | $\mathbf{0.075 \pm 0.001}$ | $\mathbf{0.071 \pm 0.001}$ | $0.169 \pm 0.004$ | $\mathbf{0.135 \pm 0.005}$ | $0.133 \pm 0.005$ |
| | **Ours (AD)** | $0.113 \pm 0.001$ | $0.071 \pm 0.001$ | $0.066 \pm 0.001$ | $0.164 \pm 0.004$ | $0.135 \pm 0.005$ | $\mathbf{0.132 \pm 0.006}$ |

Table 4: **Quantiative comparison on 1D Heisenberg model (L=30).**

periodic boundary condition, across various methods to investigate how much the methods could accurately capture the phase transition dynamics of TFIM (L=10). While LinearFM and StatisticalFM fail to accurately capture the phase transition (abrupt change of derivative), DirichletFM and our spherical and AD flow succeed in accurately estimating them.

### 4.2 ANTI-FERROMAGNETIC HEISENBERG MODEL

We consider the Hamiltonian of 1D spin-$\frac{1}{2}$ anti-ferromagnetic Heisenberg model of length $L$ with periodic boundary condition:

$$H_{\text{Heisenberg}}(\vec{c}) = \sum_{i=1}^{L} c_i(\sigma_i^x \sigma_{i+1}^x + \sigma_i^y \sigma_{i+1}^y + \sigma_i^z \sigma_{i+1}^z), \tag{12}$$

where $\sigma_i^\alpha$ are the Pauli matrices on site $i$ and the $c_i$ are individual coupling constants per bond. The Heisenberg model provides a minimal yet nontrivial Hamiltonian that captures essential quantum many-body phenomena such as interesting correlations, entanglement, and phase transitions, while still being tractable for theory and experiment.

**Learning ground states of Heisenberg model.** In table 4, similarly as in section 4.1, we report the RMSE of correlation matrix and entanglement entropy estimation. Interestingly, our Spherical flow consistently achieves the lowest RMSE for both observables and AD flow provides advantage in the class of probability path-based methods. Refer to section D for detailed experimental settings.

**Learning quantum dynamics of Heisenberg model.** While our earlier experiments focused on learning ground states for a specific class of Hamiltonians, the method itself is agnostic to this choice: it only requires empirical observations (shadows) sampled from the quantum states of interest. To demonstrate that the approach is not limited to learning ground states, we evaluate whether it can capture real time quantum dynamics under the Heisenberg Hamiltonian. Specifically, we evaluate on extrapolation task: we train the models with shadows of $|\psi(t)\rangle = e^{-iH(\vec{c})t}|0\rangle$, $t \in [0, 1)$ and use them to infer the evolved state at unseen time points $t \in [1, 2)$. In table 5, we compare our method against several baseline models on this task.

### 4.3 2-DIMENSIONAL ANTI-FERROMAGNETIC HEISENBERG MODEL.

Exploring whether the generative model can learn the ground state of 2-dimensional Hamiltonian, which becomes quickly intractable even using DMRG, would be an interesting direction. We con-

| CATEGORY | METHOD | RMSE (CORRELATION) ↓ | | | RMSE (ENTROPY) ↓ | | |
|---|---|---|---|---|---|---|---|
| | | 1k | 10k | 100k | 1k | 10k | 100k |
| Classical | RBFK | 0.293 ± 0.010 | | | 0.330 ± 0.020 | | |
| | NTK | 0.495 ± 0.008 | | | 0.306 ± 0.018 | | |
| CFM | LinearFM | 0.120 ± 0.007 | 0.091 ± 0.006 | 0.087 ± 0.006 | 0.190 ± 0.009 | 0.151 ± 0.009 | 0.145 ± 0.007 |
| | Diff-LM | 0.144 ± 0.006 | 0.107 ± 0.005 | 0.105 ± 0.004 | 0.220 ± 0.008 | 0.194 ± 0.009 | 0.192 ± 0.007 |
| CS-DFM | StatisticalFM | 0.150 ± 0.010 | 0.094 ± 0.004 | 0.090 ± 0.003 | 0.224 ± 0.006 | 0.195 ± 0.005 | 0.191 ± 0.003 |
| | **Ours (Spherical)** | **0.099 ± 0.005** | **0.073 ± 0.004** | **0.070 ± 0.004** | **0.195 ± 0.006** | **0.179 ± 0.003** | **0.177 ± 0.003** |
| | **Ours (AD)** | 0.110 ± 0.004 | 0.082 ± 0.002 | 0.080 ± 0.001 | 0.389 ± 0.014 | 0.302 ± 0.011 | 0.288 ± 0.012 |

Table 5: **Quantitative comparison on learning real time evolution of quantum state under 1D Heisenberg model Hamiltonian (L=10).**

| CATEGORY | METHOD | RMSE (CORRELATION) ↓ | | | RMSE (ENTROPY) ↓ | | |
|---|---|---|---|---|---|---|---|
| | | 1k | 10k | 100k | 1k | 10k | 100k |
| Classical | RBFK | 0.063 ± 0.002 | | | 0.131 ± 0.005 | | |
| | NTK | 0.056 ± 0.002 | | | 0.098 ± 0.002 | | |
| CFM | LinearFM | 0.120 ± 0.002 | 0.093 ± 0.002 | 0.090 ± 0.002 | 0.190 ± 0.005 | 0.161 ± 0.004 | 0.159 ± 0.003 |
| | Diff-LM | 0.168 ± 0.004 | 0.133 ± 0.002 | 0.130 ± 0.002 | 0.220 ± 0.006 | 0.185 ± 0.005 | 0.184 ± 0.004 |
| CS-DFM | StatisticalFM | 0.130 ± 0.002 | 0.105 ± 0.002 | 0.102 ± 0.002 | 0.184 ± 0.005 | 0.163 ± 0.005 | 0.160 ± 0.005 |
| | **Ours (Spherical)** | **0.090 ± 0.004** | **0.077 ± 0.002** | **0.074 ± 0.002** | 0.158 ± 0.005 | 0.123 ± 0.005 | 0.118 ± 0.004 |
| | **Ours (AD)** | 0.105 ± 0.004 | 0.077 ± 0.002 | 0.075 ± 0.002 | **0.150 ± 0.004** | **0.114 ± 0.004** | **0.112 ± 0.003** |

Table 6: **Quantitative comparison on 2D Heisenberg model (L=4x4).**

sider the following Hamiltonian: $H_{\text{2D-Heisenberg}}(\vec{c}) = \sum_{\langle i,j \rangle} c_{ij}(\sigma_i^x \sigma_j^x + \sigma_i^y \sigma_j^y + \sigma_i^z \sigma_j^z)$. In table 6, we compare our method in the task of predicting observables of 2D Heisenberg model's ground states.

## 4.4 EFFECT OF TRAINING SAMPLE SIZE ON OBSERVABLE ESTIMATION

In fig. 5, we report RMSE of correlation function with respect to various training sample size $M_{\text{train}} \in \{250, 500, 1000, 2000, 4000\}$, which is the number of shadows per Hamiltonian, on seen Hamiltonians. We fixed $M_{\text{infer}} = 100k$ across all experiments which is sufficiently large that errors are dominated by the generative model bias and not the variance over a finite number of shadows. Across the various training size, while the baselines improve only marginally, our methods achieve the lowest error and exhibits superior scaling with training samples, matching the same scaling as the exact method while attaining higher accuracy.

## 4.5 USING TETRAHEDRAL POVM SHADOWS

Pauli shadows are amongst the easiest shadows to measure experimentally compared to SIC-POVMs such as tetrahedral POVM, as one doesn't require any additional circuit depth and experimentally measuring Pauli observables is common and usually tractable. To further elaborate on the importance of Pauli-6 POVMs, for tetrahedral POVMs, physically implementing this with a quantum circuit would require an extra ancilla for each qubit, requiring total L extra ancillas.

| Method | RMSE (corr.) ↓ |
|---|---|
| LinearFM | 0.0645 ± 0.0002 |
| StatisticalFM | 0.0549 ± 0.0003 |
| **Ours (Spherical)** | 0.0539 ± 0.0002 |
| **Ours (AD)** | **0.0536 ± 0.0002** |

Table 7: Learning 1D Heisenberg model with tetrahedral POVM shadows.

Nonetheless, our approach leverages geometry arising from Pauli geometry effectively but does not strictly rely upon it. In table 7, to empirically demonstrate this, using tetrahedral POVMs, we conducted experiments learning ground states of 1D Heisenberg model (L=10). The result shows our method's efficacy beyond Pauli-6 POVM shadows.

## 5 RELATED WORK

**Shadow Representation and Generative Modeling.** Classical shadow tomography (Huang et al., 2020) introduces a powerful framework for efficiently estimating observables from randomized sin-

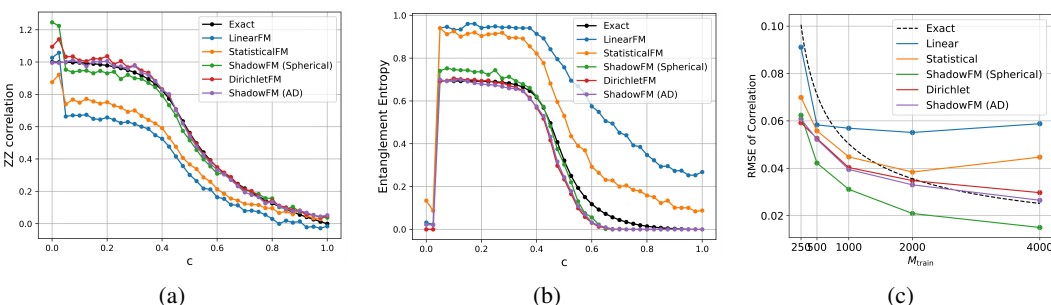

Figure 5: **(a), (b) Qualitative comparison on estimating phase transition dynamics of TFIM (L=10). (c) Effect of scaling amount of training data on Heisenberg model (L=10).**

gle qubit measurements. Extensions and improvements on classical shadows have also been proposed, including locally-biased classical shadows (Hadfield et al., 2022), shadow estimation under noise (Koh & Grewal, 2022).

In parallel, recent literature has introduced generative modeling approaches to quantum state estimation, aiming to learn distribution over the shadows. Autoregressive models (Carrasquilla et al., 2019), energy-based models (Jayakumar et al., 2024) and normalizing flows (Cranmer et al., 2019) have been explored for modeling quantum states and its corresponding shadows. However, these approaches focus on learning single quantum states corresponding to a single Hamiltonian, whereas our work addresses the problem of learning a class of multiple Hamiltonians through Hamiltonian-conditional generative modeling. In the direction of conditional generative modeling, autoregressive models (Yao & You, 2024) and diffusion models (Tang et al., 2025) have shown promising results by learning conditional likelihood and continuous diffusion processes on Euclidean space, respectively.

**Discrete Flow Matching and Riemannian Flow Matching.** There has been a recent interest in developing Flow Matching that is tailored to a specific data type or embedding space geometry. Notably, the Fisher information metric based Statistical Flow Matching (Cheng et al., 2024; 2025; Davis et al., 2024), and Dirichlet Flow Matching (Stark et al., 2024) have demonstrated strong performance in modeling discrete data, such as DNA sequences, by explicitly respecting simplex geometry of discrete probability distributions. Furthermore, Riemannian Flow Matching (RFM) (Chen & Lipman, 2023a; Arvanitidis et al., 2023) generalizes the concept to manifolds with non-Euclidean geometry. Employing DFMs and RFMs to model discrete shadows within a non-Euclidean geometry is a natural approach, and it is reasonable to expect this framework to confer advantages in generative performance.

## 6 CONCLUSION

In this work, we propose a novel generative modeling framework for predicting the ground state of a given Hamiltonian by learning the distribution of shadows, with geometric Flow Matching, motivated by the Bloch sphere geometry and pairing structure inherent in shadows. To the best of our knowledge, our approach is the first to explicitly account for the geometry of shadows and leverage geometric generative modeling for learning quantum states.

For limitations, although Flow Matching approaches have demonstrated scalable and competitive performance, it remains unclear whether they can consistently match or surpass autoregressive methods. Moreover, Anisotropic Dirichlet flow requires a pre-computations of conditional velocity field involving the computation of integrals, which introduces additional overhead at the initial stage of inference. Finally, we believe that extending our framework to more complex quantum systems, such as other 2D models constitute promising directions for future work.

## ETHICS STATEMENT

Our work presents a computational framework for learning quantum many-body ground states, which raises no ethical concerns. The methods are purely theoretical and computational, involving no human subjects or personal data, with applications limited to fundamental physics research.

## REPRODUCIBILITY STATEMENT

We provide detailed experimental setup and hyperparameters in Section D. Complete algorithmic descriptions of our shadow data generation process and geometric flow matching modifications are given in the main text. All synthetic quantum datasets are generated using standard procedures detailed in Algorithm 1 and the shadows were used to estimate physical observables as detailed in Algorithm 2. Source code will be made available upon acceptance.

## USE OF LLMS

Large language models were used solely to assist with grammar and phrasing during manuscript preparation.

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

APPENDIX

# A CLASSICAL SHADOW REPRESENTATION

---

**Algorithm 1** Measuring Shadows

**Input:** $B$ copies of unknown $n$-qubit state $\rho$, $\mathcal{U} = \{X, Y, Z\}$
**Output:** List of shadows $\mathcal{S}$
1: Initialize empty list $\mathcal{S}$
2: **for** $i = 1$ to $B$ **do**
3:     **for** $k = 1$ to $n$ **do**
4:         Sample $l_i^{(k)} \sim \text{Cat}(1/3, 1/3, 1/3)$
5:         $U_i^{(k)} \leftarrow \mathcal{U}[l_i^{(k)}]$
6:     **end for**
7:     $U_i \leftarrow U_i^{(1)} \otimes \cdots \otimes U_i^{(n)}$
8:     $l_i \leftarrow (l_i^{(1)}, \ldots, l_i^{(n)})$
9:     Measure $U_i \rho U_i^\dagger$, obtain $b_i \in \{0, 1\}^n$
10:    Append $(l_i, b_i)$ to $\mathcal{S}$
11: **end for**
12: **return** $\mathcal{S}$

---

**Algorithm 2** Observable Estimation

**Input:** List of shadows $\mathcal{S} = \{(l_i, b_i)\}_{i=1}^B$ of a n-qubit state, observable $O$, $\mathcal{U} = \{X, Y, Z\}$
**Output:** Estimation of the expectation of the observable $\langle \hat{O} \rangle$
1: **for** $i = 1$ to $B$ **do**
2:     **for** $k = 1$ to $n$ **do**
3:         $U_i^{(k)} \leftarrow \mathcal{U}[l_i^{(k)}]$
4:     **end for**
5:     $U_i \leftarrow U_i^{(1)} \otimes \cdots \otimes U_i^{(n)}$
6:     $\hat{\rho}_i \leftarrow 3 U_i^\dagger |b_i\rangle \langle b_i| U_i - \mathbb{I}$
7: **end for**
8: $\tilde{\rho} \leftarrow \frac{1}{B} \sum_{i=1}^B \hat{\rho}_i$
9: $\langle \hat{O} \rangle \leftarrow \text{Tr}(O\tilde{\rho})$
10: **return** $\langle \hat{O} \rangle$

---

**Protocol.** Let $b_i^{(k)} \in \{0, 1\}$ denote the measurement outcome on the $i$th qubit, and $U_i^{(k)}$ the unitary mapping that maps the chosen Pauli basis to the computational basis. Then, we can define the local estimator for the $i$th qubit: $\hat{\rho}_i^{(k)} = 3 U_i^{(k)\dagger} |b_i^{(k)}\rangle \langle b_i^{(k)}| U_i^{(k)} - \mathbb{I}$. The global estimator can then be constructed by taking the tensor product of the local estimators: $\hat{\rho}^{(k)} = \bigotimes_{i=1}^N \hat{\rho}_i^{(k)}$. This shadow representation significantly reduces the computational overhead while still providing unbiased estimators for local observables. With $M$ measurements of the shadow, one can reconstruct the density matrix as $\hat{\rho} = \frac{1}{M} \sum_{k=1}^M \hat{\rho}^{(k)}$, and estimate the expectation value of observable as $\langle \hat{O} \rangle = \frac{1}{M} \sum_{k=1}^M \text{Tr}(O\hat{\rho}^{(k)})$.

**Efficiency of Shadows.** A key advantage of classical shadows is their favorably efficient sample complexity. Suppose $O$ is an observable with Pauli rank $r$ (i.e., the number of non-zero coefficients when decomposed into Pauli strings). Then, to estimate $\langle O \rangle$ within additive error $\epsilon$ and failure probability $\delta$, it suffices to use $M = \mathcal{O}(r \log(1/\delta)/\epsilon^2)$ shadow measurements Huang et al. (2020). For example, pairwise correlations that we are using as an evaluation metric in this work have low Pauli rank and are thus efficiently able to be estimated.

**Example of Estimating Local Observables.** As an example, consider estimating the expectation of two-qubit observable $O = Z_1 Z_2$ from classical shadows. For each $k$-th shot, we measure all qubits independently in random Pauli bases. We only use shadows where qubits 1 and 2 are both measured in the $Z$ basis, i.e., $U_1^{(k)} = U_2^{(k)} = \mathbb{I}$. Given the measurement outcomes $b_1^{(k)}, b_2^{(k)} \in \{0, 1\}$ on those qubits, we compute:

$$\hat{O}^{(k)} \equiv \text{Tr}(O\hat{\rho}^{(k)}) = \text{Tr}(Z_1 Z_2 \hat{\rho}_1^{(k)} \otimes \hat{\rho}_2^{(k)}) = 3(-1)^{b_1^{(k)}} \cdot 3(-1)^{b_1^{(k)}} = 9(-1)^{b_1^{(k)} + b_2^{(k)}} \tag{13}$$

as an unbiased single-shot estimator for $\langle Z_1 Z_2 \rangle$. This factor of 9 arises from the definition of single-qubit inverse map used in classical shadows: $\hat{\rho}_i^{(k)} = 3|b_i^{(k)}\rangle \langle b_i^{(k)}| - \mathbb{I}$, which yields $\text{Tr}(Z_i \hat{\rho}_i^{(k)}) = 3(-1)^{b_i^{(k)}}$.

The final estimate is given by averaging the single shot estimator $\hat{O}^{(k)}$ over the $M'$ shots that satisfy the basis condition: $\langle O \rangle = \mathbb{E}[\text{Tr}(O\hat{\rho}^{(k')})] \approx \frac{1}{M'} \sum_{k'=1}^{M'} \hat{O}^{(k')}$, $k'$ denotes the indices of snapshots where qubits 1 and 2 are both measured in the Z basis.

# B    DERIVATION OF THE CONDITIONAL VELOCITY FIELD OF THE ANISOTROPIC DIRICHLET FLOW

In this section, we derive in full detail the time-dependent conditional velocity field $u_t : \Delta^{K-1} \to T\Delta^{K-1}$ such that it satisfies the continuity equation $\partial_t p_t + \nabla \cdot (p_t\, u_t) = 0$, for the pre-defined anisotropic probability path:

$$p_t(x|x_1 = e_i) = \frac{1}{Z(t)} \prod_{j=1}^{K} x_j^{\alpha_j(t)-1}, \quad \alpha_i(t) = 1 + t, \ \alpha_{\bar{i}}(t) = 1 - \gamma t, \ \alpha_j(t) = 1 \ (j \notin \{i, \bar{i}\}),$$
(14)

where,

$$Z(t) = \frac{\prod_{j=1}^{K} \Gamma(\alpha_j(t))}{\Gamma(S(t))}, \quad \alpha_\Sigma(t) = \sum_{j=1}^{K} \alpha_j(t) = K + (1 - \gamma)t.$$
(15)

Note that $\bar{i}$ denotes the conjugate pair of $i$. This will indicate the conjugate relation of the shadow measurement (i.e. $\sigma^+ \leftrightarrow \sigma^-$, $\sigma \in X, Y, Z$). In our implementation, we set $\{(i, \bar{i})\} \in \{(0, 1), (2, 3), (4, 5)\}$, which represents 3 conjugate pairs corresponding to each pauli matrix $X, Y, Z$.

Eq. 14 is a generalized Dirichlet probability path which we call *cross polytope probability path*. We incorporate anisotropy to reflect the geometry motivated from the Bloch sphere representation of shadows. Solving the continuity equation for this conditional probability path will give us the conditional velocity field $u_t$ such that generates the given $p_t$.

**Solving the Continuity Equation.**    Only $\alpha_i, \alpha_{\bar{i}}$ depend on $t$, so

$$\partial_t \log p_t = \dot{\alpha}_i \log x_i + \dot{\alpha}_{\bar{i}} \log x_{\bar{i}} - \partial_t \log Z(t) = -\gamma \log x_i + \log x_{\bar{i}} - \big[-\gamma\, \mathbb{E}_{p_t}[\log X_i] + \mathbb{E}_{p_t}[\log X_{\bar{i}}]\big],$$
(16)

hence

$$\partial_t p_t = p_t\big[-\gamma(\log x_i - \mathbb{E}_{p_t}[\log X_i]) + (\log x_{\bar{i}} - \mathbb{E}_{p_t}[\log X_{\bar{i}}])\big].$$
(17)

We seek a velocity field that solves the continuity equation Eq. 17, that acts only on coordinates $i, \bar{i}$, tangent to the simplex, and vanishing on the boundary. Motivated from Stark et al. (2024), we posit the following ansatz for the velocity field:

$$u_t(x|x_1 = e_i) = C(x_i, t)\,(e_i - x) - \gamma D(x_{\bar{i}}, t)\,(e_{\bar{i}} - x).$$
(18)

If we set $F_j(x) = f_j(x_j)\,(e_j - x)$, one finds

$$\nabla \cdot (p_t F_j) = p_t\Big[-\alpha_\Sigma f_j(x_j) + \frac{\alpha_j - 1}{x_j} f_j(x_j) - (x_j - 1) f_j'(x_j)\Big], \quad \alpha_\Sigma = \sum_{k=1}^{K} \alpha_k(t).$$
(19)

Dividing by $p_t$ gives

$$\frac{\nabla \cdot (p_t F_j)}{p_t} = -\alpha_\Sigma f_j(x_j) + \frac{\alpha_j - 1}{x_j} f_j(x_j) - (x_j - 1) f_j'(x_j).$$
(20)

Matching this to $-\partial_t \log p_t$ for $j = i, \bar{i}$ yields

$$(x_j - 1) f_j'(x_j) + \Big[\alpha_\Sigma - \frac{\alpha_j - 1}{x_j}\Big] f_j(x_j) = \log x_j - \mathbb{E}_{p_t}[\log X_j].$$
(21)

This is an ODE problem for solving $f_j(x_j)$.

**Solving the ODE problem.** Write the ODE as $f' + P(x_j)f = Q(x_j)$ by dividing by $x_j - 1$. Then,

$$P(x_j) = \frac{\alpha_\Sigma - \frac{\alpha_j - 1}{x_j}}{x_j - 1} = -\frac{\alpha_\Sigma}{1 - x_j} + \frac{\alpha_j - 1}{x_j(1 - x_j)}. \tag{22}$$

Defining the integrating factor as,

$$\mu_j(x_j) \equiv \exp\left(\int P(x_j)\,dx_j\right) = x_j^{\alpha_j - 1}(1 - x_j)^{\alpha_\Sigma - \alpha_j}, \tag{23}$$

we can derive the unique solution vanishing at $x_j = 0$:

$$f_j(x_j) = \frac{1}{\mu_j(x_j)} \int_0^{x_j} \mu_j(s) \frac{\log s - \mathbb{E}_{p_t}[\log X_j]}{1 - s}\,ds. \tag{24}$$

Identify $C(x_i, t) = f_i(x_i)$, $D(x_{\bar{i}}, t) = f_{\bar{i}}(x_{\bar{i}})$. Note that the expectation value of $\log X_j$ with respect to $p_t$ can be calculated as $\mathbb{E}_{p_t}[\log X_j] = \psi(\alpha_j) - \psi(\alpha_\Sigma)$, where $\psi(z) = \frac{d}{dz}\log\Gamma(z)$ is the digamma function.

**Alternative form via incomplete Beta function.** Here, we show the connection of Eq. 24 to the formulation of conditional velocity field in Stark et al. (2024). Define the incomplete Beta function as $I_x(a, b) = B_x(a, b)/B(a, b) = \int_0^x s^{a-1}(1-s)^{b-1} / \int_0^1 s^{a-1}(1-s)^{b-1}$. Then, one can show

$$C(x_i, t) = -\frac{B(t+1, K-1)}{x_i^t(1-x)^{K-1-\gamma t}} \partial_a I_{x_i}(a, K-1-\gamma t)\big|_{a=t+1}, \tag{25}$$

$$D(x_{\bar{i}}, t) = -\frac{B(1-\gamma t, K-1+t)}{x_i^{-\gamma t}(1-x)^{K-1+t}} \partial_a I_{x_{\bar{i}}}(a, K-1+t)\big|_{a=1-\gamma t}, \tag{26}$$

where the derivative of the incomplete Beta function with respect to the first argument is

$$\partial_a I_x(a, b) = \frac{1}{B(a, b)} \int_0^x [s^{a-1}(1-s)^{b-1}\log s - I_x(a, b)(\psi(a) - \psi(a+b))]ds. \tag{27}$$

If $\gamma = 0$, then $C(x_i, t) = -\frac{B(t+1, K-1)}{(1-x_i)^{K-1}x_i^t}\partial_a I_{x_i}(a, K-1)\big|_{a=t+1}$ and Eq. 18 becomes $u_t(x|x_1 = e_i) = C(x_i, t)(e_i - x)$. Hence this is a generalized Dirichlet flow that is consistent with the result from Stark et al. (2024).

Each $f_j(x_j) = O(x_j)$ as $x_j \to 0$ or $x_j \to 1$, so $u_t$ vanishes linearly at the boundary of simplex. Hence, our derived conditional velocity field $u_t(x|x_1 = e_i)$ is a valid vector field that generates our pre-defined probability path $p_t(x|x_1 = e_i)$.

## C  EVALUATION METRICS

For evaluation metrics, we focus on estimating the two-point correlation function and two-site Rényi-2 entanglement entropy. The correlation between two distinct qubits at sites $i$ and $j$ is defined by the expectation value of the operator $C_{ij} = \text{tr}(O_{ij}\,\rho)$, where $O_{ij} = \frac{1}{3}(X_iX_j + Y_iY_j + Z_iZ_j)$. The generated shadows are converted, via classical shadow post-processing (section 2.1), into a set of local estimators $\hat{\rho}_i^{(m)} = 3|s_i^{(m)}\rangle\langle s_i^{(m)}| - 1, i \in \{1, \ldots, L\}, m \in \{1, \ldots, M_{\text{infer}}\}$, where $M_{\text{infer}}$ denotes the number of inferred shadows. Then, the predicted correlation between qubits $i$ and $j$ is then computed as:

$$\hat{C}_{ij} = \frac{1}{3 M_{\text{infer}}} \sum_{o \in \{x, y, z\}} \sum_{m=1}^{M_{\text{infer}}} \text{tr}(\sigma_i^o \otimes \sigma_j^o \hat{\rho}_i^{(m)} \otimes \hat{\rho}_j^{(m)}), \tag{28}$$

where in TFIM, we only consider ZZ correlation, *i.e.*, $o = z$

The Rényi-2 entanglement entropy is defined as $E_{ij} = -\log(\text{tr}(\rho_{ij}^2))$, where $i, j$ denotes the index of qubit. Using shadows, we then estimate this by:

$$\hat{E}_{ij} = \frac{1}{M_{\text{infer}}(M_{\text{infer}} - 1)} \sum_{m=1,n=1,m\neq n}^{M_{\text{infer}}} \text{tr}(\mathbf{S}_{ij}^{(m,n)} \hat{\rho}_i^{(m)} \otimes \hat{\rho}_j^{(n)}), \tag{29}$$

where $\mathbf{S}_{ij}^{(m,n)}$ denotes the swap operator acting only on the subsystem consisting of the $i$-th qubit of copy $m$ and the $j$-th qubit of copy $n$.

## D  EXPERIMENTAL SETTINGS

We train all methods on a dataset constructed from 100 Hamiltonians, where the Hamiltonian parameters $\vec{c}$ is randomly sampled from $U([0, 1])$ in the TFIM case, or from $U([0, 1]^{L-1})$ in the Heisenberg model case, with $L$ denoting the number of qubits. For each Hamiltonian, we compute the ground state via exact diagonalization or DMRG and generate classical shadow samples using randomized Pauli measurements and train on them. We train on $M_{\text{train}} = 1000$ shadows per Hamiltonian. The validation and test sets each consist of 20 and 100 Hamiltonians unseen during training, respectively.

All models are trained using the Adam optimizer with a learning rate of $2 \times 10^{-3}$, a batch size of 1024, for 2000 epochs. We select the model that achieves the lowest RMSE on the validation set with respect to the correlation observable using $M_{\text{infer}} = 10k$ and then measure the RMSE on the test data. Number of integration steps used in flow model and diffusion model inference is 10. Final test performance is evaluated on both correlation and entanglement entropy. For probability path-based methods, we fixed the maximum timestep used in flow matching as $t_{\text{max}} = 4$.

The exploited model architecture for velocity field predictor is multi-head attention based Transformer with AdaLN conditioning module, hidden dimension 256. In implementing Hamiltonian-conditional velocity field predictor, we employ trainable conditional embedding layer and fixed positional embedding layer to encode positional information.

All experiments are conducted on a single NVIDIA A100 GPU (80GB). Training converges within 3 hours to 24 hours, depending on the size of the given quantum system and the chosen number of training data.

