# OpenReview forum: "ShadowFM: Geometric Approaches for Learning Quantum Many-Body States with Flow Matching on Classical Shadows"
_ICLR.cc/2026/Conference — Submitted to ICLR 2026_

### Official Review · Reviewer_L8i5 · 2025-10-17

**Soundness:** 3
**Presentation:** 2
**Contribution:** 2
**Rating:** 2
**Confidence:** 3

**Summary:**

This paper introduces ShadowFM, a generative framework for learning quantum ground states from their classical shadows. The core idea is to apply geometric flow matching on non-Euclidean manifolds inspired by the Bloch sphere, rather than using standard Euclidean approaches. By respecting the intrinsic geometry of quantum measurements, the proposed methods, Spherical Flow and Anisotropic Dirichlet Flow, compared to vanilla flow matching, achieve more accurate predictions of physical observables like correlation functions.

**Strengths:**

The paper introduces a novel framework by applying geometric flow matching to learn classical shadows of quantum states. It thoughtfully motivates this approach by connecting the non-Euclidean structure of quantum measurements to the Bloch sphere, a departure from prior works that assumed Euclidean geometry.

The proposed "Spherical Flow" and "Anisotropic Dirichlet Flow" methods are empirically validated on the Transverse-Field Ising and Heisenberg models.

**Weaknesses:**

1. The major weakness is the paper's structure and writing. A significant portion of the early sections is dedicated to explaining well-established concepts, which may be inefficient for an expert audience at ICLR conference.
- Sections 2.1, 2.2, 2.3 and Section 3 provide lengthy introductions to Classical Shadows and flow Matching. While context is necessary, these concepts are foundational and could likely be summarized more concisely, perhaps by focusing only on the specific aspects directly built upon by the authors. This would free up valuable space to elaborate on the novel contributions. If the author desires to present more specific or self-contained content, detailed information (including Algorithm 1 and 2) can be placed in the appendix.
- Furthermore, the motivation in Section 4.1 is good, but the preceding three pages of background and related works dilute the paper's focus and delay the reader's engagement with the key ideas. In a highly competitive venue, it's crucial to present the core innovation as early and clearly as possible.

A more effective structure might have been to briefly introduce the necessary concepts from FM and classical shadows within the introduction or a much shorter, combined background section, and then move directly to the motivation and detailed methodology of ShadowFM.

2. The paper's core methodological sections (4.2 and 4.3) blend established theory with the authors' modifications. It would be better if the authors delineate their novel technical contributions from the baseline frameworks of RFM and DFM. As presented, it is difficult to isolate the exact innovations beyond the application of existing tools to a new domain.

3. For the Anisotropic Dirichlet Flow, the increased methodological complexity does not consistently yield superior performance over the simpler Spherical Flow (e.g., in the Heisenberg model results of Table 3). What is the justification for this more complex model if its empirical advantage is not universally demonstrated across the tested problems?

4. A critical observation from Table 1 is that the learning-free Classical Shadow baseline consistently outperforms all trained generative models, especially on the correlation metric and in the high-sample regime. Could the authors explain this performance gap and justify the practical utility of the generative approach if it fails to surpass a direct, learning-free estimation method?

4. The primary motivation for introducing a non-Euclidean geometry is an experiment (Figure 2) suggesting that "spin errors" (e.g., $|X^{+}\rangle \rightarrow |X^{-}\rangle$) are significantly more detrimental than "basis errors" (e.g., $|X^{+}\rangle \rightarrow |Y^{\pm}\rangle \text{or} |Z^{\pm}\rangle$). However, the provided plot does not strongly support this claim. The performance gap between the two error types appears marginal across the tested error rates. The paper's assertion that spin errors are "significantly higher"  seems overstated. This weakens the central premise that a geometry specifically designed to maximize the distance between spin-flipped states is necessary and justifies the added model complexity.

5. The paper's experiments compare the proposed methods primarily against other flow-matching variants. This comparison is too narrow and fails to benchmark against the true state-of-the-art in generative modeling for quantum states. Crucially, there is no comparison against machine learning models such as [1,2,3] and a benchmark [4], which have shown strong performance on similar tasks.

[1] Huang H Y, Kueng R, Torlai G, et al. Provably efficient machine learning for quantum many-body problems[J]. Science, 2022, 377(6613).

[2] Wang H, Weber M, Izaac J, et al. Predicting properties of quantum systems with conditional generative models[J]. arXiv preprint arXiv:2211.16943, 2022.

[3] Yao J, You Y Z. ShadowGPT: Learning to Solve Quantum Many-Body Problems from Randomized Measurements[J]. arXiv preprint arXiv:2411.03285, 2024.

[4] Zhao Y, Zhang C, Du Y. Rethink the Role of Deep Learning towards Large-scale Quantum Systems[C]. Forty-second International Conference on Machine Learning, 2025.

**Questions:**

The selection of "Generative Models" as the primary area suggests the paper's main contribution lies in advancing the fundamental methodology of generative modeling. While the paper does introduce novel geometric adaptations to flow matching, these innovations are exclusively motivated, designed for, and validated on the specific problem of learning quantum states. This raises a crucial question about the work's intended contribution and audience. The work might be more appropriately positioned within an area like "Applications to Physics".

---

> ### Author Response · Authors · 2025-12-03
>
> We sincerely appreciate the reviewer's constructive feedback, mainly regarding the presentation and lack of baseline comparison. These comments indeed greatly helped us to improve our paper. We provided detailed answers and additional experimental results suggested by the reivewer below.
>
> ___
>
> __1. Reorganizing the sections and presentation__
>
> We greatly appreciate the reviewer’s feedback. Following the suggestion, we have substantially compressed Sections 2.1–2.3 and Section 3, keeping only the components directly required for understanding our method. All detailed explanations, algorithms, and extended related work have been moved to the appendix. This restructuring brings the motivation of our method earlier in the paper, and we hope this improves the clarity.
>
> In the updated manuscript, Sections 4.2 and 4.3 now explicitly distinguish our contributions from the standard RFM/FM for discrete data formulations. Each subsection first presents our method, followed by a clearly highlighted explanation of how it departs from or extends the existing flow-matching frameworks. We believe this makes the technical novelty more transparent.
>
> ___
>
> __2. Reporting accuracy of estimate from classical shadow as an oracle baseline__
>
> Classical Shadows [A] is not a generative model; it is an oracle-style estimation protocol that directly samples shadow measurements from the true test quantum states by exactly measuring shadows of a given Hamiltonian. This inherently provides the lowest variance estimator for shadow based reconstruction and shows the theoretical lower bound imposed by the inherent statistical nature of shadow tomography. In contrast our method learns distributions of training shadows and are evaluated on unseen test states by generating shadows conditioned on unseen Hamiltonian. We include this baseline primarily as a reference point for the unattainable “oracle” performance, not as a competing generative approach.
>
> ___
>
> __3. Clarification of the primary motivation to exploit geometry of shadows & Updated illustration__
>
> To clarify the result of this experiment, we updated fig. 2-(b), (c) to plot relative error of observable predictions rather than the rmse. This is done on ground states of two systems: 1D Heisenberg (L=6) and 1D TFIM (L=6). We hope this helps better elucidate the difference as it was previously unclear what the overall scale of the observable was. Even for a moderately small shadow error rate, the error composed solely of spin errors produces nearly half the error induced by basis-only errors (e.g. in Heisenberg, error rate =0.1, the relative error drops by 38% $\rightarrow$ 19%), which clearly shows how the reconstruction accuracies are significantly different.
>
> ___
>
> __4. Additional experiment with suggested baselines & argument about focusing on non-autoregressive models__
>
> Following the reviewer’s suggestion, we compare with [1] and added RBFK and NTK results to tab. 1–6. These values were obtained by solving Kernel Ridge Regression based on radial basis function kernel and neural tangent kernel, respectively, following [1]. We also implemented the diffusion-based method following [B] and included it as an additional baseline in tab. 1-6. While autoregressive (AR) models ([2], [3]) represent an interesting direction at the intersection of generative modeling and quantum many-body physics, our work focuses on developing non-autoregressive models, which offer particular advantages for long-sequence generation. For this reason, we do not include AR models as a directly comparable baseline. [4] is mainly a benchmarking paper which compares various classical ML models and AR models and claims classical ML methods can outperform deep learning approaches. Because our work focuses on developing non-AR generative model based on geometry argument, we believe this is out of our scope.
>
> [A] Huang, Hsin-Yuan, Richard Kueng, and John Preskill. "Predicting many properties of a quantum system from very few measurements." Nature Physics 16.10 (2020): 1050-1057.
>
> [B] Tang, Yehui, Mabiao Long, and Junchi Yan. "QuaDiM: A Conditional Diffusion Model For Quantum State Property Estimation." The Thirteenth International Conference on Learning Representations.

---

### Official Review · Reviewer_Fo5i · 2025-10-20

**Soundness:** 2
**Presentation:** 3
**Contribution:** 2
**Rating:** 2
**Confidence:** 3

**Summary:**

The paper introduces Shadow Flow Matching, a geometric and generative framework for modeling quantum measurement data. Rather than treating measurement outcomes as raw values, the authors embed them into a cross-polytope manifold that captures the symmetry and combinatorial structure of Pauli measurements, essentially a high-dimensional, geometry-aware analogue of the Bloch sphere. Within this space, they apply flow matching to learn distributions of outcomes conditioned on Hamiltonian parameters, allowing the model to follow the curved geometry of quantum shadows instead of a flat Euclidean one. This leads to smoother sampling and better reconstruction of observables like correlation matrices and entanglement measures, even under noise or limited data. By blending geometric insight with generative modeling, the work moves beyond treating quantum data as structureless, aligning machine learning more closely with the intrinsic geometry of quantum mechanics.

**Strengths:**

1. Mapping quantum measurements into a cross-polytope space is an original and elegant idea—it captures the symmetries of Pauli measurements while moving beyond standard Euclidean embeddings.

2. The paper nicely links discrete measurement data with continuous geometric flows, bringing flow-based generative modeling into quantum ML in a fresh and promising way.

3. The framework shows solid potential for tasks like shadow tomography, expectation reconstruction, and entropy estimation, where structure-aware modeling really helps.

4. The theory and context are well presented, clearly explaining why a geometric approach makes sense and how it connects to broader machine learning and quantum ideas.

5. The visualization of the cross-polytope flow is particularly interesting, helping to convey the intuition behind how flow trajectories respect geometric constraints and how they differ from flat-space generative flows.

**Weaknesses:**

1. Despite its novelty, the scope of practical applications appears somewhat narrow at this stage. The framework’s advantages are demonstrated primarily on specific shadow-based tasks, and it remains uncertain how easily the approach can scale to broader or more complex domains of quantum learning.

2. A current limitation is the method’s dependence on the structure of Pauli shadows. Since the model architecture and training procedure hinge on these symmetries, adapting it to non-Pauli or arbitrary measurement settings could require substantial reworking of the geometric foundation.

3. The empirical evaluation is somewhat underdeveloped. Comparisons are made to only one baseline, and while qualitative results are encouraging, they leave open questions about quantitative robustness and competitiveness against state-of-the-art generative quantum models.

**Questions:**

1. Are the current experiments confined to the 1D anti-ferromagnetic Heisenberg model, or has the framework been tested on other many-body systems?

2. How does the proposed approach compare to diffusion-based quantum models, such as Generative Quantum Machine Learning via Denoising Diffusion Probabilistic Models?

3. Given the method’s dependence on Pauli-based shadows, could the framework extend to alternative measurement protocols like? Is the model’s success contingent on Pauli-specific symmetry?

4. Could this geometric-flow framework be repurposed for broader quantum modeling tasks—for instance, simulating quantum dynamics?

---

> ### Author Response · Authors · 2025-12-03
>
> We sincerely appreciate the reviewer's constructive feedback, mainly regarding the lack of experiments and baseline comparison. We provided detailed answers and additional experimental results suggested by the reivewer below.
>
> ___
>
> __1. Additional Experiments on 2D systems and larger Heisenberg model__
>
> We thank the reviewer for highlighting the importance of evaluating beyond 1D systems. Following this suggestion, we have now conducted experiments on a 2D Heisenberg model (L=4x4). The new 2D results confirm that our method can scale beyond the classically tractable 1D regime and is superior over other non-autoregressive baselines.
>
> Also, following the reviewer’s suggestion, we trained on the 1D Heisenberg model with L = 30, and the method continues to outperform baseline approaches, confirming its favorable scaling behavior.
>
> ___
>
> __2. Comparison with diffusion-based methods__
>
> QuDDPM learns the distribution of quantum states in an unsupervised manner, while our work targets Hamiltonian-conditional quantum state generation using a conditional generative model. As such, we respectfully argue that the results are not directly comparable. In addition, QuDDPM employs parametrized quantum circuits, making classical simulation intractable for even moderately large systems (e.g., L=30).
>
> We also emphasize that flow matching with stochastic paths (v-prediction) formally subsumes diffusion models (ε-prediction). Following the reviewer’s recommendation, we replicated the diffusion-based approach (Diff-LM [A]) described in QuaDim [B] and incorporated its results into Tab. 1–6.
>
> [A] Li, Xiang, et al. "Diffusion-lm improves controllable text generation." Advances in neural information processing systems 35 (2022): 4328-4343.
> [B] Tang, Yehui, Mabiao Long, and Junchi Yan. ""QuaDiM: A Conditional Diffusion Model For Quantum State Property Estimation."" The Thirteenth International Conference on Learning Representations.
>
> ___
>
> __3. Why Pauli-based shadows are important__
>
> We believe our method is about obtaining advantages in geometric modeling asymmetric IC-POVMS such as Pauli-6 POVMs. Pauli-6 POVMs are a widely used measurement and can be much more easily physically implemented compared to other SIC-POVMs such as tetrahedral POVMs. Due to the geometry of pauli eigenstates, they have their anti pairs, such as X- to X+. Our method proposes a way to incorporate this asymmetric geometry of the discrete data space and a more appropriate flow that reflects this geometry. That being said, SIC-POVMs do not require this consideration of asymmetry and the use of FM for discrete data including our sphericla flow would be still effective. However, our ADFM incorporating asymmetric geometry would not gain meaningful advantage over ordinary simplex FMs in this case. To empirically demonstrate this, we conducted experiments using tetrahedral POVMs, which is described in tab. 7.
>
> To further elaborate on the importance of Pauli-6 POVMs, for tetrahedral POVMs, physically implementing this with a quantum circuit would require an extra ancilla for each qubit, requiring total L extra ancillas. Then, one would have to apply specific two-qubit unitaries for appropriate basis transform and measure in the computational basis. This overhead motivates experimentalists to choose pauli-6 shadows over tetrahedral shadows. For instance, if the state is coming from an experiment such as an optical lattice, there is no straightforward way to include these ancilla and introduce the necessary rotations but measuring the Pauli shadows is straightforward. Moreover, theories regarding the Pauli-6 shadows (e.g. estimation efficiency) are much more prevalent in the quantum information and computing community. This could be some of the reasons why development of ML methods that are tailored to Pauli-6 shadows is desirable.
>
> ___
>
> __4. Additional experiment on learning quantum dynamics__
>
> We agree with the reviewer that our framework is not restricted on learning ground state-specific properties, and is easily extendable to learning arbitrary quantum state distribution. Our method is a generic geometric flow based generative model applicable to any probability distribution over classical shadows. To demonstrate this, we have now included experiments on real-time evolution of quantum states under 1D TFIM with L = 10, where the target distribution corresponds to the state at multiple time points.

---

### Official Review · Reviewer_vYzA · 2025-10-31

**Soundness:** 3
**Presentation:** 2
**Contribution:** 3
**Rating:** 6
**Confidence:** 4

**Summary:**

The paper presents Shadow FM, which is a flow based method to generate Pauli POVM measurements of a quantum ground state. The Hamiltonian of the system can be used as a conditioning input to generate these samples. There are two approaches in the paper: (i) a spherical / Riemannian flow aligned with the Bloch-sphere geometry and (ii) an anisotropic Dirichlet probability path. The authors test their methods on Heisenberg and TFIM chains.

**Strengths:**

The paper is well writing and the flow matching framework that the paper uses is well motivated by the geometry of the quantum states. Being a non-autoregressive method is also a positive,  as the shadow distribution is unlikely to have 1D nature for interesting quantum states. The experiments are thorough and interesting.

**Weaknesses:**

1. All the experiments presented are for 1D models. This is a concerning as the ground states of 1D models have efficient classical representation in terms of MPS representations. 2D experiments would have substantially improved the quality of the results

2. Motivation of restricting to ground states of Hamiltonians is unclear to me? Why not thermal states or states produced by real time evolution? I don't see the learning task itself using any information about the fact this is a ground state of a Hamiltonian.

**Questions:**

1. For learning ground states, could this method be enhanced by adding a variational component to the loss that also tries to minimize the estimated energy of the state?

2. What is the main bottleneck faced by the authors to go to 2D experiments?

3. In the phase transition studies, do the authors observe any changes in the behavior of the learning algorithm across the phase transition?

4. How do the authors ensure that the distribution over shadow states that the learned model samples from for a new Hamiltonian actually correspond to a physically allowed quantum state? Can a projection step be built into the inference pipeline to project to physically allowed states?

---

> ### Author Response · Authors · 2025-12-03
>
> We sincerely appreciate the reviewer's constructive feedback, mainly regarding the lack of experiments and baseline comparison. We provided detailed answers and additional experimental results suggested by the reivewer below.
>
> ___
>
> __1. Additional Experiments on 2D systems and larger Heisenberg model__
>
> We thank the reviewer for highlighting the importance of evaluating beyond 1D systems. Following this suggestion, we have now conducted experiments on a 2D Heisenberg model (L=4x4) (tab. 6). The new 2D results confirm that our method can scale beyond 1D systems where MPS scales well and is superior over other non-autoregressive baselines.
>
> ---
>
> __2. Additional Experiment on learning real time evolution of quantum states__
>
> We agree with the reviewer that our framework does not rely on ground state-specific properties, and is easily extendable to learning arbitrary quantum state distribution. ADFM is a generic geometric flow based generative model applicable to any probability distribution over classical shadows. To demonstrate this, we have now included experiments on real-time evolution of quantum states under 1D TFIM with L = 10, where the target distribution corresponds to the state at multiple time points.
>
> ---
>
> __3. About adding variational loss to minimize the estimated energy corresponding to shadow distribution__
>
> Thanks for the interesting question. In a separate but not yet published work, we have done exploration of trying to variationally minimize shadows. Unfortuantely, incorporating a variational component to the loss while learning the shadow distribution of  ground states will naturally lead to unphysical solutions. This is because there are distributions of shadows that do not correspond to a physical state (or more often are unlikely to have come from a physical state). Some of these unphysical distribution of shadows appear to have an energy lower than the true ground state energy if you evaluate the energy from them as if it comes from a physical state. Therefore, a loss function which incorporates a variational energy term would generally push the model away from the physically realizable manifold and cause the optimizer to find distributions of shadows that have no corresponding physical state but give an energy smaller then the true ground state energy. Successfully adding such an additional term therefore would be a highly non-trivial open question and it is out of the scope of this work.
>
> ---
>
> __4. About projecting the predicted state to the manifold of physically allowed states__
>
> There is nothing that formally prevents the distribution over shadows to correspond to a physically allowable state.  In practice, we rely on the inductive bias of the training data,  the actual shadow samples are produced from the physically valid states, which empirically keeps the learned model within the physically allowed region.  While it would be useful to project to physically allowable staes, to the best of our knowledge, there is no clean or tractable method for projecting an arbitrary shadow distribution onto the set of distributions induced by physical quantum states. Even for a single qubit, one can construct a distribution over shadows whose estimated density matrix has a negative eigenvalue or such that represent unphysical state. For example, obtaining only X+ and Z+ shadows for a single qubit, would be a simple case that corresponds to an unphysical state. Deciding whether a distribution of shadows corresponds to any valid quantum state is closely related to the quantum marginal (N-representability) problem; "Given reduced density matrices for some subsets of particles, does there exist a full many-body quantum state whose marginals match these reduced density matrices?"), which is QMA-hard.  Approximately answering this question has been a long-standing problem in quantum chemistry as it gives useful variational lower bounds.  For similar reasons, we strongly believe that there will be no simple formulation such as projection step for classical shadows which guarantees it comes from a physical state.

---

### Official Review · Reviewer_Mwxx · 2025-11-01

**Soundness:** 3
**Presentation:** 3
**Contribution:** 2
**Rating:** 4
**Confidence:** 2

**Summary:**

The paper introduces ShadowFM, a geometric flow matching framework for learning ground-state quantum many-body wavefunctions via the distribution of classical shadows.
Instead of modeling full quantum states, ShadowFM learns to generate shadow measurements—compact randomized representations of quantum states—and uses them to estimate physical observables such as correlation functions and entanglement entropies.

**Strengths:**

- Originality: the paper presented a novel framework called ShadowFM that combines low matching generative modeling with classical shadow tomography for learning quantum many-body ground states.
- Quality: The technical development looks sound and link both geometric and quantum information theory. The paper demonstrates a good understanding of both and integrates them coherently.
- Clarity: The paper is clearly written, with a well-organized structure. Figures and tables of experiments look good.
- Significance: The work study a problem that is significant in bridging geometric deep generative modeling and quantum many-body learning. It proposes a scalable, data-driven alternative to classical shadow reconstruction and autoregressive models.

**Weaknesses:**

- Computational overhead: The anisotropic Dirichlet flow requires precomputing and integrating Beta-function–based terms, which could limit practicality; runtime and memory costs are not quantified in very detail.
- Restricted empirical scope: Experiments are limited to 1D spin chains (TFIM and Heisenberg). The scalability and performance of ShadowFM on larger or higher-dimensional systems are not demonstrated.

**Questions:**

- Could the authors explain more about the scalability of their methods? For example, for Heisenberg model, could the experiments for $L=30$ be conducted? Also, what about other models?

---

> ### Author Response · Authors · 2025-12-03
>
> We appreciate the reviewer for valuable feedback. Please see the detailed answers for each question below. We hope this resolves your concern.
>
> ---
>
> __1. Computational overhead of ADFM__
>
> We appreciate the reviewer’s insightful comment regarding the computational cost of the anisotropic Dirichlet flow. We agree that ADFM requires precomputing and integrating Beta-function–based terms; however, this is only needed once, exclusively during inference, and not during training. Although we didn't find a need to do this, in pratice we can precompute a lookup table of all required Beta-integral terms, which can then be efficiently reused to accelerate inference for any downstream task. Computing this lookup table a single time takes approximately 20 minutes on an A100 (80GB) GPU. After this one-time generation, both memory and runtime overheads become negligible relative to the overall pipeline. We will clarify this point in the revised version.
>
> ---
>
> __2. Additional Experiments on 2D systems and larger Heisenberg model__
>
> We thank the reviewer for raising the question of scalability beyond the 1D spin chains presented in the main paper. In response, we have expanded our experiments in the revised manuscript:
>
> - Larger 1D Heisenberg: Following the reviewer’s suggestion, we evaluated on the 1D Heisenberg model with L = 30, and the method continues to outperform non-autoregressive baseline approaches, confirming its favorable scaling behavior (tab. 4).
> - Time dynamics: We extend our method to learn time evolution under the 1D TFIM (L = 10), illustrating that our method can handle quantum dynamics, not only the static ones such as ground states (tab. 5).
> - 2D systems: We now include results on the 2D Heisenberg model (4×4), demonstrating that our method generalizes beyond one-dimensional systems (tab. 6).

---

### Author Response · Authors · 2025-12-03
**Final comment to AC and reviewers**

We sincerely appreciate AC for handling our submission under this unusual circumstance. Below, we summarize our answers to the major questions and revision of the manuscript reflecting suggestions by the reviewers. Please see the full corresponding rebuttals for more details.


__1. Limited experimental scope - lack of 2D systems and larger models (Mwxx, vYzA, Fo5i).__

We add experiments on 1D Heisenberg model with (L=30) (tab. 4), and real-time quantum dynamics under 1D TFIM (L=10) (tab. 5), 2D Heisenberg model (L=4×4) (tab. 6), demonstrating generalizability to higher-dimensional systems, scalability in system size, and generalizability to learn arbitrary conditional distribution beyond Hamiltonian-conditional ground states, respectively.

__2. Baseline comparison is insufficient (L8i5, Fo5i).__

Following the reviewer's suggestion, we add comparisons with diffusion-based methods, and classical machine learning modles based on kernel methods (RBFK, NTK). We elaborate classical shadow oracle as a theoretical lower bound reference for clarification. Additional results are now reported in tab. 1–6.

__3. Unclear presentation of contribution & Clarifying motivation of using Bloch sphere geometry (L8i5).__

We reorganize Sections 2.1–2.3 and Section 3 by moving detailed explanations to the appendix, and restructured Sections 4.2–4.3 to explicitly distinguish our novel contributions from standard flow-matching frameworks. We also revise fig. 2(a) in order to more rigorously verify our claim about geometry by testing on both Heisenberg and TFIM and plotting with relative error to show significance in the difference between values. We hope this improves the clarifies motivation to use Bloch sphere geometry.

__4. Concerns about physical validity of learned shadow distributions (vYzA).__

We explain that there is no clean known way of projecting shadow distributions to physical states, and we rely on empirical training data's inductive bias to keep the model within physically allowed regions. In a separate unpublished work, we also explored adding variational term to minimize the energy function which was suggested by the reviewer, but we saw optimizer push the distribution of shadows to an unphysical state. Hence, this is an open question without trivial solution and we believe it is out of this work's scope.

__5. Importance of Pauli-6 POVMs over other measurement schemes (Fo5i).__

We empirically demonstrate efficacy with tetrahedral POVMs in tab. 7 and show our method can generalize to other POVM shadows. We also explain that our work specifically focuses on Pauli-6 POVMs because it is significantly easier to implement physically, making our geometry-aware approach practically important. We further argue that restricting our generative modeling to the Pauli-6 POVM is physically well-motivated and incurs no loss of generality and the Pauli-6 setting offers a natural foundation due to its well-developed, extensive theoretical literature.

__6. Computational overhead of ADFM (Mwxx).__

We clarifiy that Beta-function terms are precomputed only once during inference (not training), taking ~20 minutes on an A100 GPU, after which memory and runtime overheads become negligible.

---

### Meta-Review · Area_Chair_yX2c · 2025-12-25

**Summary:**

This paper studied how to learn quantum many-body states with flow matching on classical shadows, and along this two methods, namely Riemannian-based method and probability path-based method are applied.

The reviewers have some general concerns about the results:
- Limited experimental scope - lack of larger systems and 2D systems;
- Baseline comparison is insufficient;
- Other technical points, such as motivation of using Bloch sphere geometry, physical validity of learned shadow distributions, Pauli-6 POVMs over other measurement schemes, etc.

The authors revised the paper carefully for addressing these points. In particular:
-  Experiments were added on 1D Heisenberg model with (L=30) (tab. 4), and real-time quantum dynamics under 1D TFIM (L=10) (tab. 5), 2D Heisenberg model (L=4×4) (tab. 6).
- Baseline comparisons were added with diffusion-based methods, and classical machine learning modles based on kernel methods (RBFK, NTK).
- Other various details were clarified.

Given the circumstance this year, the AC carefully checked both reviews as well as the authors' rebuttal and updated manuscript. It is considered that the revised manuscript was better than the previous version, but it's probably not enough to meet the bar of ICLR. Some concerns remained here, and the AC found some further concerns that were not adequately discussed:
- It's not very clear why the specific classical shadow of quantum many-body state setting is of general interest to the ICLR audience. Reviewer Fo5i raised the point that "The framework’s advantages are demonstrated primarily on specific shadow-based tasks", which the AC agrees. The authors explained why studying Pauli-6 POVMs, but in any case this is still a very physics setting, and it's not that clear how this can be extended to more scenarios of learning quantum problems or AI for physics as a whole.
- The AC agrees with Reviewer Mwxx's concern about computational overhead and scalability. In the reply the authors mentioned that for the studied cases it takes ~20 minutes on an A100 GPU, but it's not very clear if this is convincing enough for the efficiency - this is not a very small computational cost (around 10^3 seconds on a top-level GPU) while the problem scale is also not that large (especially under the shadow setting). It would be more convincing if the cost is comprehensively studied under different scales and systems, or would be perfect if an explicit complexity upper bound is proved.

In all, given the existing concerns and the overall 3 negative initial scores by the reviewers, the decision is rejection for ICLR 2026.

**Reviewer Concerns:**

The authors did a reasonable job to address reviewers' concerns, and many points are addressed. But there are still points outstanding - see the metareview.

**Reviewer Scores:**

I would say that the scores will marginally increase if the reviewers were able to participate fully in the discussion, but I don't think it will overturn the 3 negative initial reviews to all positive.

---

### Decision · Program_Chairs · 2026-01-26

Reject